# Persuasive Features that Drive the Adoption of a Fitness Application and the Moderating Effect of Age and Gender

**Kiemute Oyibo \*** and **Julita Vassileva**

Multi-User Adaptive Distributed Mobile And Ubiquitous Computing (MADMUC) Lab, Department of Computer Science, University of Saskatchewan, Saskatoon, SK S7N 5C9, Canada; jiv@cs.usask.ca

\* Correspondence: kiemute.oyibo@usask.ca

**Abstract:** Fitness apps equipped with various persuasive features have become popular worldwide due to the physical inactivity crisis. However, there is a limited understanding of the most important persuasive features that drive their adoption and the moderating effect of age and gender. To bridge this gap, we designed storyboards illustrating six of the commonly employed persuasive strategies in persuasive health applications: Goal-Setting/Self-Monitoring, Reward, Social Learning, Social Comparison, Competition and Cooperation. We conducted an empirical study in which we asked the participants to evaluate their receptiveness to the six persuasive features and their intention to use a fitness app that features them. The result of our Partial Least Square Path Modeling (PLSPM) shows that, overall, Goal-Setting/Self-Monitoring is the strongest predictor of the *intention to use* a fitness app, followed by Reward and Competition, both of which are in second place. However, Social Learning and Social Comparison turn out to be non-predictors of *intention to use*. Based on these findings, we recommend that a minimally viable (one-size-fits-all) fitness app, in a personal setting, should support a Goal-Setting/Self-Monitoring feature, coupled with a Reward feature, to increase its appeal to a wide audience. Moreover, in a social setting, it should support a Competition feature to increase its appeal to a wide audience. We discuss these findings and the gender and age differences in the relationships between users' receptiveness to the six persuasive features and their intention to use a fitness app that support them.

**Keywords:** persuasive technology; persuasive features; fitness app; intention to use; goal-setting; self-monitoring; competition; cooperation; age; gender

## 1. Introduction

Owing to the ever-increasing prevalence of physical inactivity and its attendant effects, fitness apps are becoming more and more popular and essential in our daily lives. In particular, fitness applications that support users in various settings, including at home, have become useful given the coronavirus pandemic, which has confined people to their homes, thereby increasing sedentary lifestyles such as playing video games and watching movies on the Internet. Given that many people can no longer go to the gym to workout with their personal trainers or others in social settings due to gym closures, the need for fitness applications in physical personal setting and virtual social setting as a tool for becoming fit physically and mentally cannot be overstressed. Usually, they are equipped with a number of persuasive features to motivate and facilitate the desired behavior change. Some of the commonly employed persuasive features in fitness apps on the market today include Goal-Setting/Self-Monitoring, Reward, Social Learning, Social Comparison, Competition and Cooperation [1–4]. Research [5–7] has shown that these persuasive features have the potential of promoting the target behavior change in a

personal and/or social context. However, there are limited studies focused on uncovering how well the perceived persuasiveness of these persuasive features predicts users' intention to use fitness apps to motivate their behavior change. Specifically, research on the relationship between users' receptiveness to persuasive features and their intention to use fitness apps that support them is scarce. To bridge this gap, we designed a questionnaire of storyboards illustrating six of the commonly employed persuasive features in persuasive health applications (PHAs): Goal-Setting/Self-Monitoring, Reward, Social Learning, Social Comparison, Competition and Cooperation. Thereafter, in an online survey, we asked the study participants residing in North America to evaluate their receptiveness to each persuasive feature illustrated on a storyboard and their intention to use a fitness app prototype called "HOMEX," which is aimed to motivate regular exercise in a home setting. The study, which employed a mixed-method approach, aims to support designers of persuasive fitness applications by providing insight into the key persuasive features that drive the adoption of fitness applications on the market by potential users.

The result of our PLSPM shows that, overall, Goal-Setting/Self-Monitoring ($\beta = 0.52$, $p < 0.001$) is the strongest predictor of the *intention to use* the fitness app, followed by Reward ($\beta = 0.17$, $p < 0.01$), Competition ($\beta = 0.17$, $p < 0.05$) and Cooperation ($\beta = 0.13$, $p < 0.01$). However, Social Learning and Social Comparison turned out to be non-predictors of *intention to use*. Comparatively, the relationship between Cooperation and *intention to use* is significantly stronger for females than for males, while the relationship between Goal-Setting/Self-Monitoring and *intention to use* is significantly stronger for older users than younger users. Overall, the findings reveal that personal features are stronger predictors of the adoption of a fitness app than socially oriented features. Specifically, our results show that among users residing in North America (which are mostly individualists by country of origin), Goal-Setting/Self-Monitoring is the strongest motivator of the *intention to use* a fitness app. However, older people are more likely to adopt a fitness app based on its Goal-Setting/Self-Monitoring feature than younger people. Secondly, females are more likely to adopt a fitness app based on its Cooperation feature than males. Thirdly, while Social Comparison is a motivator of the adoption of a fitness app for younger people, it is a demotivator for older people. Based on our overall findings, we recommend that designers of fitness apps, especially for the North American audience, should support Goal-Setting and Self-Monitoring (accompanied by Reward) in minimally viable apps to increase their appeal to a wider audience and chances of being used. Further, in a social context, fitness app designers should support persuasive features such as Competition and Cooperation (in additional to the personal features implemented on a group-basis) to foster intrinsic motivation [8–10] and accountability [2], respectively. These social features have the potential of increasing user engagement in the target behavior among socially oriented users.

The rest of the paper is organized as follows. Section 2 focuses on the background on commonly employed persuasive features in fitness apps and related work. Section 3 describes the research method. Section 4 presents the results of our path analysis. Section 5 dwells on the discussion of our findings. Finally, Section 6 concludes the paper.

## 2. Background and Related Work

In this section, we present an overview of the six persuasive features of a PHA we investigated and the related work.

### 2.1. Definition of Persuasive Features

Persuasive features are supportive/motivational features with which PHAs are equipped to increase their perceived persuasiveness and actual effectiveness in changing behavior. Table 1 shows all six persuasive features addressed in this paper and their definitions. The six persuasive features, which are commonly employed in persuasive applications in the health domain [2,11], were adopted from Oinas-Kukkonen and Harjumaa's [7] persuasive system design (PSD) model. For example, Reward, which entails offering incentives to users such as points, levels and badges, has been widely

studied in prior studies [4]. However, most of the studies such as [12–14] have been focused on users' receptiveness. Munson and Consolvo [4] particularly investigated the effectiveness of Reward in a real-life fitness app. They found that reward elements such as trophies and ribbons were not successful in motivating most of the study participants. According to the authors, this "*raises questions about how such rewards should be designed*." We argue that, prior to knowing how Reward should be designed, it will be useful for designers to know *ab initio* whether Reward as a persuasive strategy has the potential to drive the adoption of fitness applications. Secondly, Goal-Setting/Self-Monitoring, which is aimed at setting a goal and tracking users' performance towards achieving the set goal, has been widely studied as well. For example, Munson and Consolvo [4] investigated the effectiveness of Goal-Setting as a persuasive strategy in a real-life fitness application. They found that setting primary and secondary weekly goals was beneficial to the study participants. Moreover, in a qualitative study, Orji et al. [15] found that "*Self-monitoring is the cornerstone of many health and wellness persuasive interventions*" (p. 1). However, both persuasive features have not been studied as a possible predictor of the adoption of a fitness application. Finally, just as Reward and Goal-Setting/Self-Monitoring, social features such as Cooperation, Social Comparison, Competition and Social Learning have been studied as well in the prior literature. However, most of the existing studies [2,13,16] focus on users' receptiveness to these social strategies and not their potential, in the face of other persuasive features, to predict users' adoption of fitness applications.

**Table 1.** Persuasive features and definition [17–19].

| Feature | Definition of Feature |
| --- | --- |
| Goal-Setting/Self-Monitoring | A persuasive feature that allows users to set goals and track their performance over time. |
| Reward | A persuasive feature that allows incentives to be awarded to users for the accomplishment of their goal. |
| Cooperation | A persuasive feature that allows users to work together to achieve a collective goal and reward upon reaching their goal. |
| Competition | A persuasive feature that allows users with a common goal and mutually exclusive reward to compete with one another to attain them. |
| Social Comparison | A persuasive feature that allows users to view and compare their performance and achievements with those of others. |
| Social Learning | A persuasive feature that allows users to observe the behaviors and achievements of other users and respond accordingly. |

### 2.2. Gender and Age Differences

Research shows that demographic variables could be employed to segment populations for the purpose of personalization of persuasive strategies to the target audience. In particular, gender and age have been commonly employed to segment populations and studied in the literature [20,21]. Oyibo et al. [5] found that gender influences the receptiveness of users to persuasive strategies. Specifically, in a non-domain-specific context, the authors found that males are more likely to be receptive to Reward and Competition than females. Similarly, in the physical activity domain, Van Uffelen et al. [22] found that males are more likely to be motivated by competitive physical activities than females. Regarding age difference, in a non-domain-specific context, Oyibo et al. [5] found that younger people (under 24 years old) are more likely to be receptive to Competition, Social Comparison and Social Learning than older people (over 24 years old). Moreover, in the energy-conservation domain, Shih and Jheng [3] found that age influences users' receptiveness to persuasive strategies. For example, they found that Reward is more persuasive to younger adults (under 40 years old) than older adults (over 41 years old). However, they found that Self-Monitoring and Cooperation are more persuasive to older adults than to younger adults. Thus, they recommended that designers should employ different persuasive strategies when targeting different age groups of users. Due to the age and gender differences in users' receptiveness to certain persuasive strategies, we became interested in

how both demographic variables moderate the relationships between commonly employed persuasive features (e.g., Goal-Setting/Self-Monitoring, Cooperation, etc.) and users' adoption of fitness apps that support them.

### 2.3. Captology and Persuasive Technology

Fogg [6], the pioneer of persuasive technology, defined "Captology" (an acronym for Computers As Persuasive Technologies) as the study of computers as persuasive technologies aimed to change attitude and behavior. Thus, "Captology," which is often used interchangeably with the terminology "Persuasive Technology," is regarded as a field of study that deals with the intersection of "Computer" and "Persuasion" as shown in Figure 1. In this light, Captology refers to the employment of computers and persuasive techniques from social psychology in the art of persuasion (motivating people, fostering compliance, changing behavior and attitude, etc.) in different fields of human endeavors. Examples of computer applications aimed at changing behavior and attitude include desktop applications (virtual agent), mobile applications (e.g., fitness apps equipped with behavior models [23,24]), social media (e.g., Facebook, etc.), persuasive health games [25–27], etc. All of these applications are regarded as persuasive technologies, which have the potential to influence user attitudes and behaviors through persuasion and/or social influence. However, there are limited studies with regard to the relationship between the persuasive features of persuasive technologies and their targeted behavioral outcomes. Specifically, very few studies have been conducted to investigate the relationship between persuasive features and the *intention to use* a persuasive application. Rather, most of the related studies (e.g., [28–31]) have been conducted in the context of TAM, which focuses primarily on the relationship between user experience (UX) design attributes (such as *perceived usefulness*, *perceived usability*, *perceive aesthetics*, etc.) and *intention to use/actual use*. In the next section, we cover a cross-section of the relevant studies, including the few studies that focused on the relationship between persuasive features and persuasive outcomes such as *intention to use/actual use* of persuasive systems.

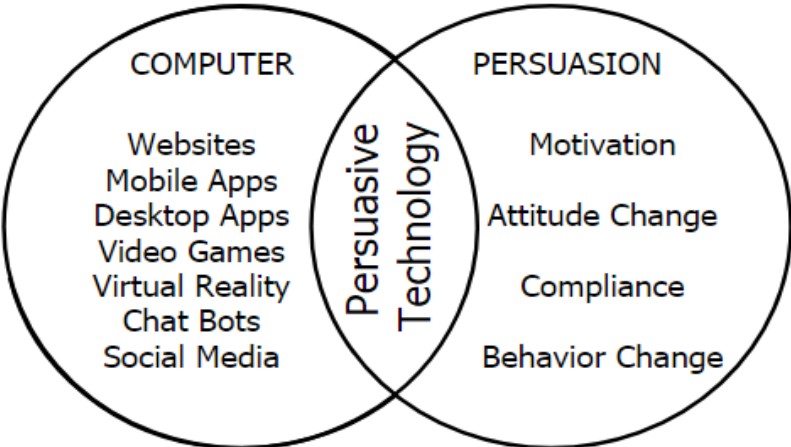

**Figure 1.** Persuasive Technology dubbed "Captology" [32].

### 2.4. Relationship between Persuasive Features and Intention to Use/Actual Use

Researchers [23,33] have adapted Bandura's [34] social cognitive model of reciprocal determinism to the persuasive technology context. The model (see Figure 2) shows that personal factors, environmental factors, and the target behavior interact with one another in a reciprocal fashion to shape behavior. The personal factors include social cognitive factors such as *self-efficacy*, *self-regulation* and *outcome expectation* [23].

The environmental factors, in the context of persuasive technology, include personal persuasive features (Goal-Setting, Self-Monitoring, etc.,) and social persuasive features (Social Learning, Cooperation, Competition, etc.). Finally, the target behavior includes behavioral outcomes such

as intention to use, engagement, behavior performance, etc. A number of studies [23,35–37] in the physical activity domain have been carried out based on this behavior change model. In this paper, we focus specifically on the relationship between system features and target behavior in the context of persuasive technology, especially for health interventions.

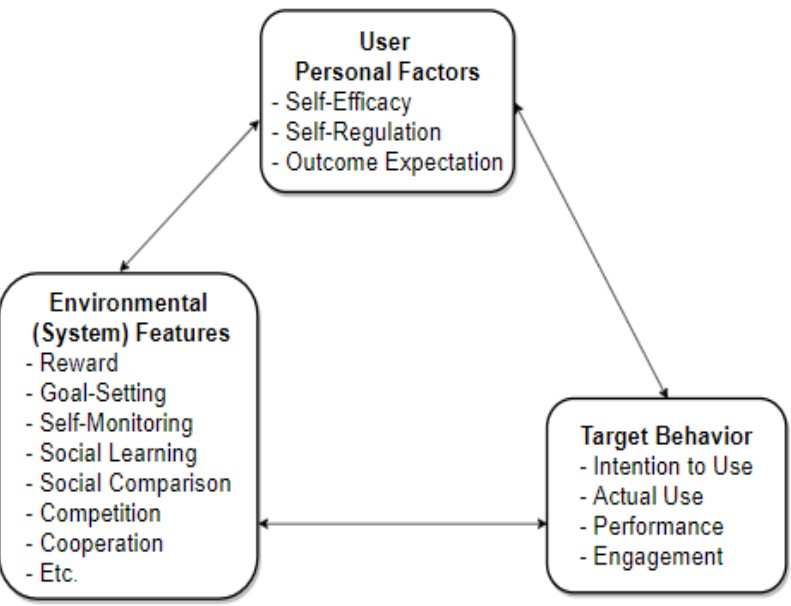

**Figure 2.** Social cognitive model of reciprocal determinism in persuasive technology context [23].

### 2.5. Related Work on Relationship between Persuasive Features and Intention to Use/Actual Use

A number of studies have been carried out related to persuasive features and their potential to change behavior. Stibe and Oinas-Kukonen [33] developed a model for the design of persuasive systems that support user engagement in collaborative interaction. The model shows the social predictors of *user engagement* and *behavior intention* in a socially oriented persuasive system integrated with Twitter. The authors found that Social Learning, Social Facilitation and Cooperation (through *perceived persuasiveness*) are significant predictors of both target constructs. Overall, *perceived persuasiveness* has the strongest total effect on *behavioral intention* and *user engagement*, followed by Social Learning, Cooperation, and Social Facilitation. However, apart from the investigated persuasive system not being a health-based system, the authors only focused on social features as possible predictors of the target constructs. In other words, they did not consider personal features such as Goal-Setting and Self-Monitoring, perhaps, because they were not relevant in the Twitter-based social system they investigated. Moreover, Lehto et al. [38] carried out an evaluation of a web-based persuasive system (which they called behavior change support system) for healthy eating and losing weight. Specifically, they investigated the factors that influence the usage of the persuasive system. They found that among the six factors they investigated, *perceived persuasiveness*, *unobtrusiveness* and *design aesthetics* are the strongest (overall) predictors of the *actual use* of the system, with the first two factors (only) having a direct influence on *intention to use*. (*Unobtrusiveness*, in particular, is a measure of how well a system fits with the environment in which the user uses it [39].) Drozd et al. [39] also carried out a similar study in the eating domain. They found that *perceived persuasiveness* and *unobtrusiveness* are the strongest predictors of the *intention to use* a persuasive system designed to promote health eating. The main limitation of Lehto et al. [38] and Drozd et al.'s [39] studies, in the light of our study, is that they treated the persuasive features of the persuasive system as a monolithic construct (*perceived persuasiveness*) rather than as different constructs such as Reward, Cooperation, etc. Thus, it is hard to tell which of the commonly employed persuasive features are the strongest predictors of the *intention to use* a persuasive system. Our current study aims to bridge this gap.

Specifically, due to the identified gaps, the aim of our study is to investigate the strongest predictors of users' intention to use a home-based fitness app that supports commonly employed persuasive features: Goal-Setting/Monitoring, Reward, Social Learning, Social Comparison, Competition and Cooperation. We used storyboards to illustrate each of these persuasive features.

## 3. Method

In this section, we present our research questions, measurement instruments for the six persuasive features we investigated and the demographic information of participants.

### 3.1. Research Questions

In our study, given the paucity of research in the area of persuasive features as predictors of fitness app use, as shown in the exploratory model in Figure 3, we set out to answer the following research questions (RQs) using an exploratory approach:

RQ1. Can persuasive features predict users' intention to use a fitness app?

RQ2. Which of the six commonly employed persuasive features is/are the strongest predictors of users' intention to use a fitness app?

RQ3. How do gender and age moderate the interrelationships among the persuasive features and users' intention to use a fitness app?

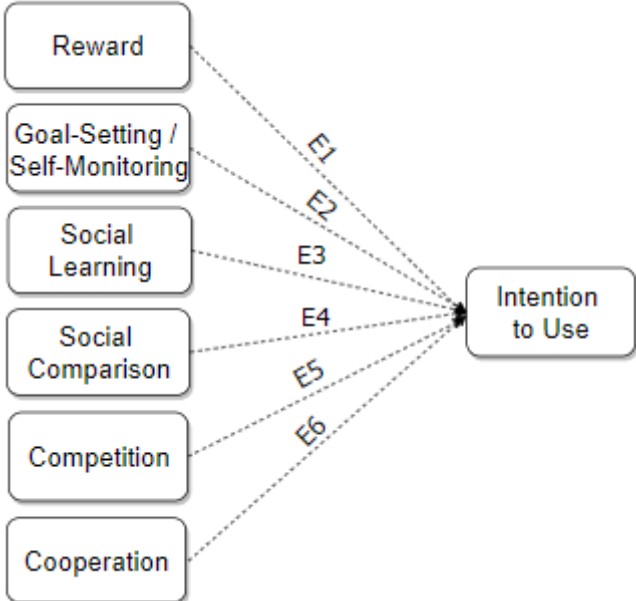

**Figure 3.** Research model of intention to use a fitness app.

### 3.2. Storyboards

To answer our research questions, we employed storyboards to illustrate each of the persuasive features of interest. Figure 4 shows the storyboard illustrating the Goal-Setting/Self-Monitoring feature. In the storyboard, the target user sets a tiny goal of 4000 calories (4 Kcal) on a given day. In our study, we want to understand how well persuasive strategies such as this would influence or motivate users to adopt (use) a fitness app aimed at encouraging physical activity. Storyboards such as this have been widely used in previous studies (e.g., [2,3,40,41]) to elicit useful feedback from potential users.

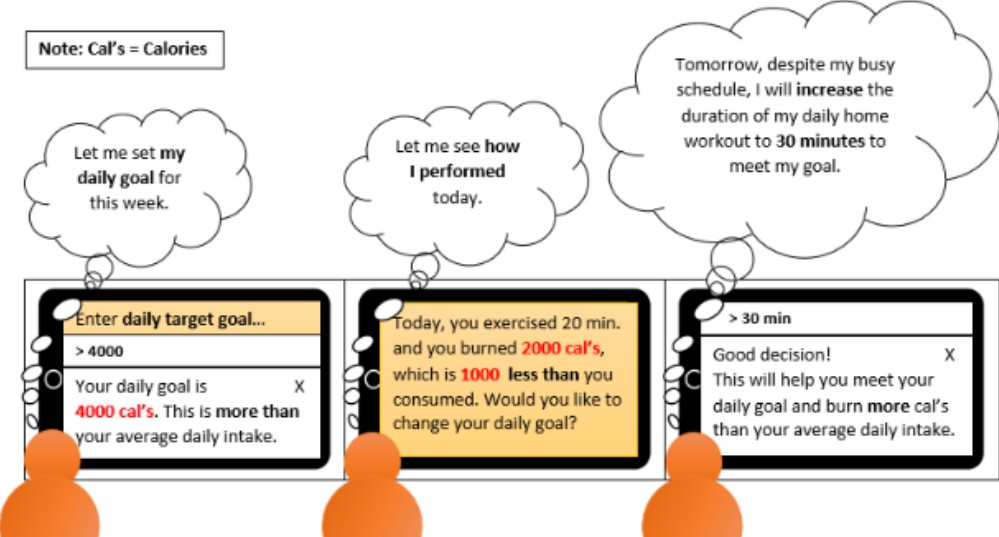

**Figure 4.** Storyboard illustrating the Goal-Setting/Self-Monitoring feature of a fitness app.

## 3.3. Measurement Instruments

We carried out an online survey to investigate users' receptiveness to the six persuasive features illustrated on the storyboards and their intention to use a fitness app in which they are featured. Prior to administering the storyboards and their corresponding questions to participants, we presented a description of a fitness app prototype (which we called the "Homex App") to set the tone for users' response to our questionnaire. The application description read as follows:

> *Imagine you want to improve your personal health and fitness level. Given the challenges (e.g., time, cost, weather, etc.) associated with going to the gym regularly, the "Homex App" has been created, say by health promoters in your neighborhood, to support your physical activity.*

Thereafter, with respect to each storyboard illustrating each of the six persuasive features (Goal-Setting/Self-Monitoring, Reward, Social Learning, Social Comparison, Competition and Cooperation), the first question in Table 2 was asked. Thereafter, the last question on *intention to use* the fitness app was asked. Particularly, we combined Goal-Setting and Self-Monitoring as one construct because we regard them as complementary. In other words, if a fitness app supports the setting of goals, then users should be automatically allowed to have the option of tracking their activities aimed at meeting the set goals as well. Otherwise, the persuasive strategy of Goal-Setting may be less effective if users cannot keep track of their progress and achievement of their set goals.

Prior to answering the questions shown in Table 2, the study participants were asked to study each storyboard, identify and choose the correct persuasive feature being illustrated from a number of options. This was done to ensure that they paid attention to and understood the persuasive feature illustrated in each storyboard prior to answering the first question shown in Table 2. We believe, in so doing, the reliability of participants' responses would be enhanced. Wrong responses to incorrectly identified storyboard's persuasive features were treated as missing data points and replaced by the respective average scores during the data analysis. Specifically, we used the adapted version of the Perceived Persuasiveness scale [39]—previously used by other studies (e.g., [23])—to measure the *perceived persuasiveness* of each feature. For *intention to use*, we used a single-item scale. Prior research [42] has shown that single-item scales could be as reliable as multi-item scales. After the participants had finished rating the storyboards in terms of their receptiveness to the illustrated persuasive features, they were requested to provide comments to justify their ratings. The question read, "*Provide comments about this application feature [persuasive strategy illustrated on the storyboard] to justify your rating here [textbox].*" This open question was included in the study in order to triangulate the quantitative with the qualitative findings.

**Table 2.** Study's constructs and indicators.

| Criterion | Overall Question and Items |
|---|---|
| Perceived Feature [23] | Imagine that you are using the Homex App presented in the storyboard above to track your physical activity, to what extent do you agree with the following statements: <br> 1. This feature of the app would influence me. <br> 2. This feature of the app would be convincing. <br> 3. This feature of the app would be personally relevant to me. <br> 4. This feature of the app would make me reconsider my physical activity. <br> 5. Provide comments about this application feature [persuasive strategy illustrated on the storyboard] to justify your rating here [textbox]. |
| Intention to Use | Assuming the app, together with the various features, described earlier on, would be available to me, I predict that I will use it. |

## 3.4. Participants

Our study was submitted to and approved by the authors' University Research Ethics Board. Thereafter, it was posted on Amazon Mechanical Turk (a crowdsourcing platform) to recruit participants residing in Canada and United States. Each of the participants was compensated with USD $1.50 for their time. Overall, 279 participants took part in the study. However, after cleaning, 228 were left (see Table 3): males (132), females (95), and unidentified (1). The majority of the excluded 51 participants was as a result of non-completion of the survey. About 72% of the valid participants had North America (Canada and United States) as their country of origin. The other demographic information about the valid participants is enumerated in Table 3.

**Table 3.** Demographics of participants (n = 228).

| Variable | Subgroup | Number | Percent |
|---|---|---|---|
| Gender | Male | 132 | 57.9 |
| | Female | 95 | 41.7 |
| | Others | 1 | 0.4 |
| Age | 18–24 | 38 | 16.7 |
| | 25–34 | 122 | 53.5 |
| | 35–34 | 45 | 19.7 |
| | 45–54 | 16 | 7.0 |
| | 54+ | 7 | 3.1 |
| Education | Technical/Trade School | 31 | 13.6 |
| | High School | 39 | 17.1 |
| | BSc | 107 | 46.9 |
| | MSc | 33 | 14.5 |
| | PhD | 6 | 2.6 |
| | Others | 2 | 0.9 |
| Country of Origin | Canada | 89 | 39.0 |
| | United States | 98 | 43.0 |
| | Others | 41 | 18.0 |
| Continent of Origin | North America | 164 | 71.9 |
| | South America | 10 | 4.4 |
| | Europe | 13 | 5.7 |
| | Africa | 11 | 4.8 |
| | Asia | 13 | 5.7 |
| | Middle East | 5 | 2.2 |
| | Others | 2 | 0.9 |

## 4. Results

This section focuses on the results of the PLSPM, carried out using R's "plspm" package [43], including the evaluation of the measurement models and the analysis of the structural models at the global and subgroup levels.

### 4.1. Measurement Models

The evaluated preconditions in each of the measurement models include indicator reliability, internal consistency reliability, convergent validity, and discriminant validity of the constructs.

Indicator Reliability. All of the indicators in the respective measurement models had an outer loading greater than 0.7.

Internal Consistency Reliability. This criterion assessed the reliability of each construct in each measurement model. It was based on the composite reliability metric (DG.rho), which was greater than 0.7.

Convergent Validity. This criterion assessed the degree to which the indicators of each construct in each measurement model was related. It was based on the Average Variance Extracted, which was greater than 0.5.

Discriminant Validity. This criterion assessed the degree to which the different constructs in each of the measurement models are unrelated. It was based on the crossloading metric. Our results showed that no indicator loaded higher on any other construct than its own construct [44].

### 4.2. Global Structural Model

Figure 5 shows the global model. The model is characterized by three parameters: goodness of fit (GOF), coefficient of determination ($R^2$) and path coefficients ($\beta$s). The GOF value captures how well the model fits its data, while the $R^2$ value represents the amount of variance of the target construct (*intention to use*) that is accounted for by the persuasive features. Finally, the $\beta$ value represents the strength of the relationship between each persuasive feature and *intention to use*. The result of the PLSPM shows that four of the six persuasive features significantly predict the *intention to use* a fitness app, with the overall model accounting for 64% of its variance. Goal-Setting/Self-Monitoring ($\beta = 0.52$, $p < 0.001$) turns out to be the strongest predictor, followed by Reward ($\beta = 0.17$, $p < 0.01$), Competition ($\beta = 0.17$, $p < 0.05$) and Cooperation ($\beta = 0.13$, $p < 0.01$). Unfortunately, at the global level, Social Leaning and Social Learning have no significant effect on *intention to use*.

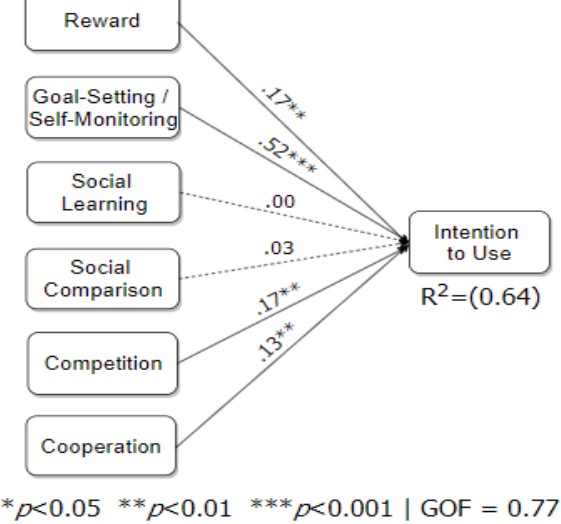

**Figure 5.** Global model of intention to use a fitness app.

*4.3. Gender-based Structural Models*

Figure 6 shows the gender- and age-based submodels. The respective submodels are similar to the global model regarding some of the relationships. For example, as in the global model, Goal-Setting/Self-Monitoring has the strongest relationship with *intention to use*: for male (β = 0.59, *p* < 0.001), for female (β = 0.43, *p* < 0.001), for younger people (β = 0.45, *p* < 0.001) and for older people (β = 0.58, *p* < 0.001). However, the results of our multigroup analyses showed that there is gender as well as age difference regarding some of the relationships. In the gender-based model, the relationship concerning Cooperation is significantly different for both genders (*p* < 0.05). It is significant for females (β = 0.27, *p* < 0.001), but non-significant for males (β = 0.04, *p* = n.s). Moreover, in the age-based model, the relationship concerning Social Comparison is significantly different for both age groups (*p* < 0.05). It is positive for younger people (β = 0.19, *p* < 0.05), but negative for older people (β = −0.19, *p* < 0.05). Finally, concerning Goal-Setting, the relationship is significantly stronger for older people (β = 0.58, *p* < 0.001) than for younger people (β = 0.45, *p* < 0.001).

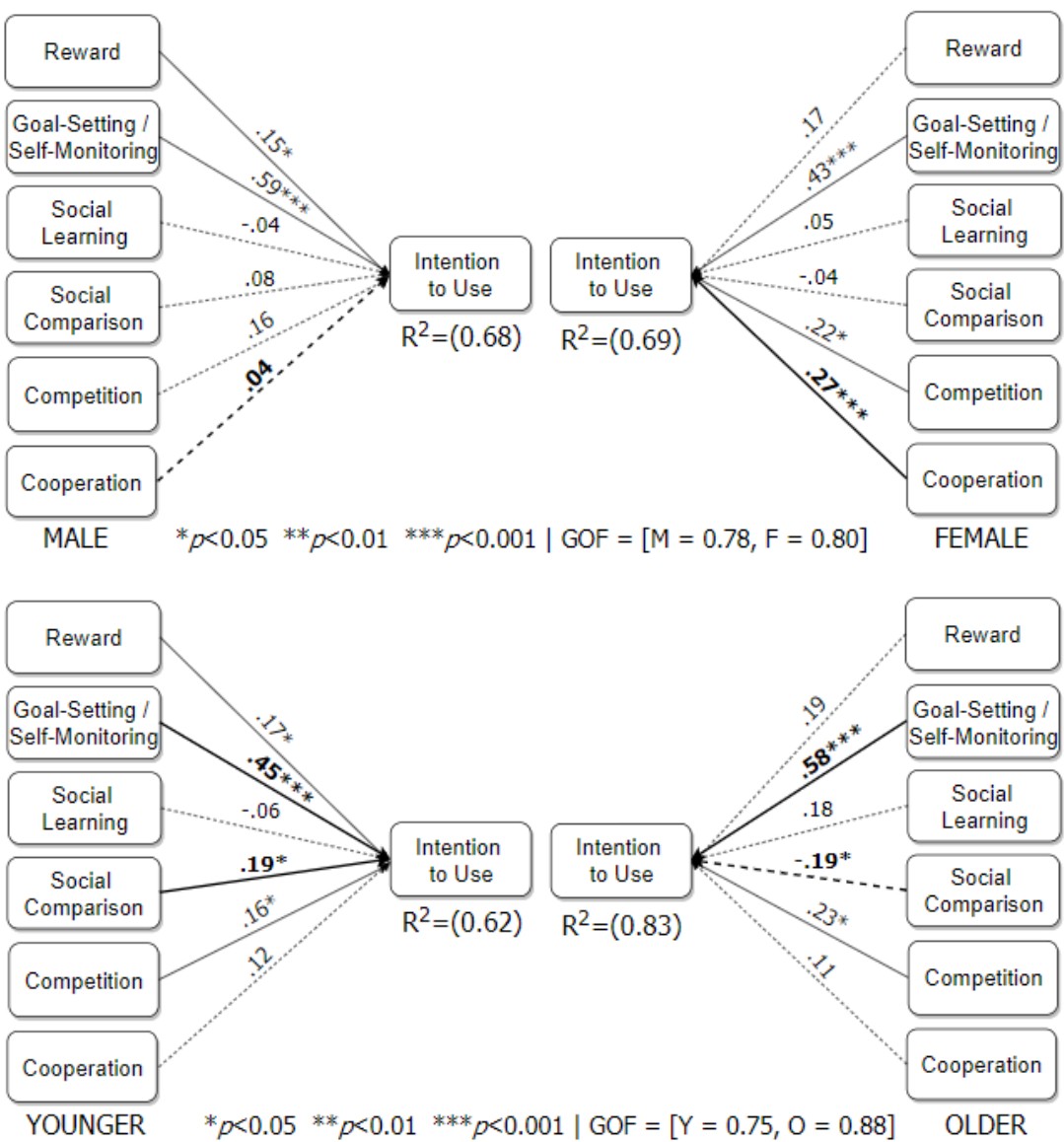

**Figure 6.** Gender-based submodels (above) and age-based submodels (below). The bold path indicates where the two subgroups in question significantly differ at *p* < 0.05. Younger people (18–34 years old) and older people (34+ years old).

*4.4. Sample of Comments Supporting the Relationship between Users' Receptiveness to Persuasive Features and Intention to Use a Fitness App*

In addition to the PLSPM, we manually went through the comments provided by the study participants to uncover qualitative evidence that supports the relationship between users' receptiveness to the significant persuasive features and *intention to use* a fitness app. Table 4 shows a cross-section of participants' comments supporting the relationship between users' receptiveness to Goal-Setting/Self-Monitoring and Reward, on one hand, and *intention to use*, on the other hand. In addition, Table 4 shows the participants' levels of receptiveness to both significant personal persuasive features in question and *intention to use*.

**Table 4.** Participants' comments on Goal-Setting/Self-Monitoring and Reward features and receptiveness profile.

| No. | Participants' Comments | Profile | Remark (PF, ITU) |
|---|---|---|---|
| P23 | *"I like the idea because it clearly tracks your calories which I usually wouldn't consider while doing exercise."* | [GST/SMT = 7, ITU = 6] | (High, High) |
| P26 | *"I just don't think I could live up to the goals and that would depress me."* | [GST/SMT = 2, ITU = 1] | (Low, Low) |
| P39 | *"Rewards often convince me to log into apps ... "* | [REWD = 5, ITU = 6] | (High, High) |
| P05 | *"I am exercising because I like it; I don't need points rewards."* | [REWD = 1, ITU = 1] | (Low, Low) |

PF = Personal Feature, GST/SMT = Goal-Setting/Self-Monitoring, REWD = Rewards, ITU = Intention to Use.

Table 5 shows a cross-section of participants' comments supporting the relationship between users' receptiveness to Competition and Cooperation on one hand, and *intention to use*, on the other hand. In addition, Table 5 shows the participants' levels of receptiveness to both significant social persuasive features in question and *intention to use*.

**Table 5.** Participants' comments on competition and cooperation features and receptiveness profile.

| No. | Participants' Comments | Profile | Remark (SF, ITU) |
|---|---|---|---|
| P69 | *"This would motivate and influence me to push harder every day to achieve the top rank (or attempt to) therefore this level of competition does indeed convince influence motivate me and is directly relevant to myself."* | [CMPT = 6, ITU = 5] | (High, High) |
| P127 | *"I'm not the competitive type. I don't do things to be better than others."* | [CMPT = 1, ITU = 1] | (Low, Low) |
| P44 | *"Having someone else depending on my activity to gain rewards would influence me to meet my goal."* | [COOP = 6.75, ITU = 7] | (High, High) |
| P92 | *"I don't want to depend on others, and I don't want to impose on others to do exercise."* | [COOP = 1, ITU = 2] | (Low, Low) |

SF = Social Feature, CMPT = Competition, COOP = Cooperation, ITU = Intention to Use.

Table 6 shows comments supporting the moderating effect of age in the relationship between users' receptiveness to Social Comparison and *intention to use*. In addition, Table 6 shows the levels of receptiveness to Social Comparison and *intention to use* for the younger and older people.

Finally, Table 7 shows a summary of the relationship between persuasive features and *intention to use*. It is based on the results shown in Figures 5 and 6. The main takeaway from the summary is that, regardless of age and gender, users' receptiveness to Goal-Setting/Self-Monitoring has a positive influence on their intention to use a fitness app. The second takeaway is that, for females, younger and older people, users' receptiveness to Competition has a positive influence on their intention to use a fitness app as well. The other takeaways are discussed in Section 5.

**Table 6.** Participants' comments on social comparison feature and receptiveness profile.

| No. | Age | Participants' Comments | Profile | Remark (SF, ITU) |
|---|---|---|---|---|
| P12 | 18–24 | *"Similar to competitive leader boards being able to compare yourself to friends for a nice push is great."* | [SCOMP = 5, ITU = 6] | (High, High) |
| P127 | 25–34 | *"I don't like comparison/competition."* | [SCOMP = 1, ITU = 1] | (Low, Low) |
| P19 | 35–44 | *"Comparing myself to others can be harmful and demotivating. Someone always loses."* | [SCOMP = 1, ITU = 6] | (Low, High) |
| P58 | 35–44 | *"If I can see the results of other people in the group and everybody except me was successful it would make me try harder in the future. I would put more effort into reaching my goal in order to keep up with the others."* | [SCOMP = 6.25, ITU = 1] | (High, Low) |
| P06 | 54+ | *"Not interested keep my goals personal."* | [SCOMP = 2, ITU = 5] | (Low, High) |

SF = Social Feature, SCOMP = Social Comparison, ITU = Intention to Use.

**Table 7.** Summary of the relationship between persuasive features and intention to use.

| Relationship | Global | Male | Female | Young | Old |
|---|---|---|---|---|---|
| Reward | ✔ | ✔ | ✕ | ✔ | ✕ |
| Goal-Setting/ Self-Monitoring | ✔ | ✔ | ✔ | ✔ | ✔ |
| Cooperation | ✔ | ✕ | ✔ | ✕ | ✕ |
| Competition | ✔ | ✕ | ✔ | ✔ | ✔ |
| Social Comparison | ✕ | ✕ | ✕ | ✔ | − |
| Social Learning | ✕ | ✕ | ✕ | ✕ | ✕ |

✔ = Positive significant relationship with intention to use at $p < 0.05$; − = Negative significant relationship with intention to use at $p < 0.05$; ✕ = Non-significant relationship with intention to use ($p$ = n.s).

## 5. Discussion

We have presented a path model of the *intention to use* a fitness app that supports users' behavior change in the physical activity domain. The goodness of fit (GOF) of the global model is 0.77. This is considered a high value in the PLSPM community, indicating that the model fits its empirical data to a large degree. Similarly, the GOFs of the submodels are above 0.70 as well, with the submodel for older people having the highest value (0.88), while that for younger people having the lowest value (0.75). According to Hussain et al. [45], a GOF value of 0.10, 0.25, and 0.36 is an indication that the overall validation of a model by its empirical data is small, medium, and large, respectively. Moreover, the global model accounts for 64% of the variance of *intention to use*. Similarly, the submodels account for more than 60% of the variance of *intention to use*, with that for older people having the highest value (83%), while that for younger people having the lowest value (62%). Again, like the GOF, the $R^2$ values, which are above 60% for the global and submodels, are high, indicating that the models' significant predictors do well in explaining the variance of *intention to use*. According to Sanchez [43], $R^2$ values above 60% are considered high values; those between 60% and 30% are considered moderate; and those less than 30% are considered low. Particularly, in the global model, four of the investigated persuasive features (Goal-Setting/Self-Monitoring, Reward, Cooperation, and Competition) have a significant relationship with the *intention to use* a fitness app. We discuss our findings in detail in the context of the significant personal features (Goal-Setting/Self-Monitoring and Reward) and social features (Cooperation and Competition). In addition, we discuss the age and gender differences regarding relationships concerning Cooperation, Social Comparison and Goal-Setting/Self-Monitoring, on one hand, and *intention to use*, on the other hand.

### 5.1. Personal Features as Drivers of Fitness App's Use

Our PLSPM shows that at the global level (Figure 5) and subgroup levels (Figure 6), Goal-Setting/Self-Monitoring is the strongest predictor of the *intention to use* a fitness app. For example, at the global level, the path coefficient of the relationship is ($\beta$ = 0.52, $p$ < 0.001), while that at the subgroup levels is ($\beta$ > 0.40, $p$ < 0.001). These significant path coefficients suggest that, regardless of gender and age, Goal-Setting and Self-Monitoring are very important complementary persuasive features in motivating behavior change in a fitness app. Our treating Goal-Setting and Self-Monitoring as complementary features of a fitness app is supported by some of the participants' comments, in which goal-setting and tracking are mentioned side by side. For example, as shown in Table 4, P15 commented thus about the Goal-Setting/Self-Monitoring feature: "*I think it is good to see results and progression on a daily basis to set new and higher goals.*" Similarly, P18 commented thus, "*I love that I can set a daily goal and track my progress towards it in real time [.] This gives me time to adjust my exercise routine or diet to reach my goal.*" In particular, the multigroup analysis (see Figure 6) shows that the relationship between Goal-Setting/Self-Monitoring and *intention to use* is significantly stronger ($p$ < 0.05) for older people ($\beta$ = 0.58, $p$ < 0.001) than for younger people ($\beta$ = 0.45, $p$ < 0.001). This suggests that older people are more likely to adopt a fitness app based on its Goal-Setting/Self-Monitoring persuasive features than younger people. Overall, the finding that Goal-Setting/Self-Monitoring is the most important predictor of users' intention to use a fitness app corroborates Orji et al.'s [15] qualitative finding, in which the authors stated that Self-Monitoring is "*the cornerstone of many health and wellness persuasive interventions*" (p. 1).

Furthermore, we found that Reward ($\beta$ = 0.17, $p$ < 0.01) is an important feature as well in determining the *intention to use* a fitness app. Particularly, there is no significant gender and/or age difference with regard to the relationship between users' receptiveness to Reward and *intention to use* (see Figure 6). Hence, in the global model, both personal features (Goal-Setting/Self-Monitoring and Reward) are the strongest predictors of *intention to use*, with Goal-Setting/Self-Monitoring ($\beta$ = 0.52, $p$ < 0.001) being stronger than Reward. Both findings with regard to the relationships between Goal-Setting/Self-Monitoring and Reward on the one hand, and *intention to use* on the other hand, can be interpreted as follows. The higher users are receptive to the Goal-Setting/Self-Monitoring and Reward features of a fitness app, the higher is their *intention to use* the fitness app to motivate their physical activity. On the flipside, the lower users are receptive to the Goal-Setting/Self-Monitoring and Reward features, the lower is their *intention to use* the fitness app. Table 4 shows a snippet of the participants' comments and profile with regard to both personal features, which attest to these findings. For example, P23, who rated the Goal-Setting/Self-Monitoring (M = 7/7) and *intention to use* (M = 6/7) high, commented positively, "*I like the idea because it clearly tracks your calories which I usually wouldn't consider while doing exercise.*" On the other hand, P26, who rated the Goal-Setting/Self-Monitoring (M = 2/7) and *intention to use* (M = 1/7) low, commented negatively, "*I just don't think I could live up to the goals and that would depress me.*" Similarly, with respect to Reward, P39, whose average rating is relatively high (M = 5/7), rated the *intention to use* the fitness app as high as well (M = 6/7) and gave a positive comment about the persuasive feature. On the other hand, P05, whose average rating is relatively low (M = 1/7), rated the *intention to use* the fitness app as low (M = 1/7) and gave a negative comment about the persuasive feature as well.

Apart from Goal-Setting/Self-Monitoring (coupled with Reward) being a fundamental feature of a fitness app, one possible explanation for its significant influence on *intention to use* is that the studied audience is an individualist culture: citizens and/or residents of Canada and United States. People living in this type of culture are mostly independent, self-motivated and goal-driven. In prior studies based on Social Cognitive Theory, Oyibo et al. [35,46] found that Perceived Self-Efficacy and Perceived Self-Regulation (mapped to persuasive features such as Goal-Setting, Self-Monitoring, and Reward in the application domain) are the strongest determinants of physical activity behavior. While Oyibo et al.'s [35] found that, in the behavior theory domain, personal factors are the strongest determinants of behavior change for the individualist culture, in the application domain, the current

study found that Goal-Setting/Self-Monitoring is the strongest driver of users' intention to use a fitness app to motivate their behavior change.

*5.2. Social Features as Drivers of Fitness App's Use*

Apart from personal features, we found that social features such as Competition ($\beta = 0.17$, $p < 0.05$) and Cooperation ($\beta = 0.13$, $p < 0.01$) are significant predictors of the *intention to use* a fitness app. This means that the higher users are receptive to a fitness app's Competition and Cooperation features, the more likely they are to form favorable intentions to use the app. On the flipside, the lower users are receptive to both features, the less likely they are to form favorable intentions to use the app. In particular, the multigroup analysis (see Figure 6) shows that the relationship between Cooperation and *intention to use* is significantly stronger for females ($\beta = 0.27$, $p < 0.001$) than for males ($\beta = 0.04$, $p = $ n.s). This suggests that females are more likely to adopt a fitness app based on its support for the Cooperation persuasive feature than males. This finding is consistent with Van Uffelen et al.'s [22] extant finding in the physical activity domain. The authors found that women were more likely than men to spend time with others in their physical activity.

Table 5 shows a snippet of participants' comments and profile with regard to Cooperation and Competition features, which support the PLSPM findings in the global model. For example, P44, who rated Cooperation (M = 6.75/7) and *intention to use* (M = 7/7) high, commented positively, "Having someone else depending on my activity to gain rewards would influence me to meet my goal." On the other hand, P92, who rated Cooperation (M = 1/7) and *intention to use* (M = 2/7) low, commented negatively, "I don't want to depend on others, and I don't want to impose on others to do exercise." Similarly, with respect to Competition, P69, whose average rating is relatively high (M = 6/7), rated the *intention to use* the fitness app as high as well (M = 5/7) and gave a positive comment about the persuasive feature. On the other hand, P127, whose average rating of Competition is relatively low (M = 1/7), rated the *intention to use* the fitness app as low as well (M = 1/7) and gave a negative comment about the persuasive feature.

Apart from Competition and Cooperation (see Figure 6), we found that for younger people, Social Comparison has a positive relationship with *intention to use* ($\beta = 0.19$, $p < 0.05$). However, for older people, Social Comparison has a negative relationship with *intention to use* ($\beta = -0.19$, $p < 0.05$). This means that the higher (lower) younger users are receptive to a fitness app's Social Comparison feature, the more (less) likely they are to have a favorable *intention to use* the app. On the other hand, the lower (higher) older users are receptive to the Social Comparison feature, the more (less) likely they are to have a favorable *intention to use* the app. Table 6 shows sample comments about Social Comparison from the younger and older groups. For example, P127 (a younger participant who commented, "*I don't like comparison/competition*") rated Social Comparison high (M = 1/7) and *intention to use* high (M = 1/7). On the other hand, P19 (an older participant who commented, "*Comparing myself to others can be harmful and demotivating. Someone always loses*") rated Social Comparison low (M = 1/7) and *intention to use* high (M = 6/7).

Based on the above, we submit that, apart from personal features (Goal-Setting/Self-Monitoring and Reward), social features (such as Competition and Cooperation) can be employed as well to motivate fitness apps' use. In particular, for younger people, Social Comparison can be employed to motivate their adoption of a fitness app. This age-based finding can be explained by a prior finding that younger people are more likely to be receptive to social comparisons than older people. In two different empirical studies, Callan et al. [47] found that younger people reported higher levels of social comparison tendency than older people. Moreover, in the social context of persuasive technologies, Oyibo et al. [14] found that younger people are more likely to be persuaded by Social Comparison as a persuasive strategy than older people. In sum, the finding that social features (such as Competition, Cooperation, etc.) is a predictor of *intention to use* in the application domain is in line with Oyibo et al.'s [35] finding in the behavior theory domain. Specifically, in the context of Social Cognitive Theory, the authors found that Social Support (next to Self-Efficacy and Self-Regulation)—mapped to

social features such as Competition, Cooperation, etc.,—was a significant driver of physical activity behavior among people living in an individualist culture.

*5.3. Design Guidelines*

Based on our key findings summarized in Table 7, we recommend a number of persuasive technology design guidelines. In Figure 5 and Table 6, we found that Goal-Setting/Self-Monitoring, regardless of gender and age, is the strongest and most consistent predictor of the *intention to use* a fitness app aimed to motivate behavior change. Given this finding, among the persuasive features we investigated, we recommend that, in a one-size-fits-all fitness app, regardless of gender and age, Goal-Setting/Self-Monitoring should be given priority as an essential persuasive strategy for motivating behavior change in the physical activity domain. Due to its importance and effectiveness, many health apps (e.g., *Houston* [11], *UbiFit* [48], etc.) often employ Goal-Setting/Self-Monitoring as a key persuasive feature to motivate and drive physical activity behavior. Specifically, research has shown that Goal-Setting as a persuasive strategy, complemented by Self-Monitoring, will be more effective in motivating behavior change if set goals are "SMART" (Specific, Measurable, Attainable, Relevant, and Time-bound) [18]. Moreover, to increase the effectiveness of the Goal-Setting/Self-Monitoring feature in fitness apps, we recommend that the Reward feature be employed to motivate behavior change alongside Goal-Setting/Self-Monitoring. Research shows that users are more likely to meet their goals if their accomplishments are rewarded, especially immediately. According to Oyibo et al. [5], Reward has "*the tendency to provide an immediate reinforcement and present users something to work for since it is often difficult to visualize the short-term benefit of most behavior*" (p. 40). Prior research [14] has shown that people living in individualist culture are receptive to Reward as a persuasive strategy for motivating behavior change.

Apart from the personal features, we recommend that, among the social features we investigated, Competition and Cooperation should be employed to motivate our target audience (mostly individualists) to engage in physical activity behavior. Competition can be implemented with the aid of leaderboards, while Cooperation can be implemented in a way that allows users to work together in groups of two or more to set a collective goal and earn a joint reward upon achieving their goal. While Competition fosters intrinsic motivation [1], Cooperation has the potential of fostering a sense of accountability among collaborative users. Specifically, P158 commented thus about Cooperation, "*Having someone to help keep you accountable is always a good motivator*." Therefore, based on all of the findings, we suggest that, among people living in individualist cultures, to motivate behavior change in the fitness domain at the personal level, Goal-Setting, Self-Monitoring, and Reward should be employed. Moreover, at the social level, in addition to the personal features (the basic features of a fitness app [11]), Competition and Cooperation should be employed to motivate the behavior change of socially oriented individuals.

*5.4. Summary of Main Findings and Contributions*

In summary, we present the key findings of our investigation in the light of our research questions as follows:

1. Goal-Setting/Self-Monitoring feature is the strongest predictor of the *intention to use* a fitness app among people living/residing in individualist countries such as Canada and United States, followed by Reward, Competition and Cooperation.
2. Cooperation feature is more likely to motivate females to use a fitness app than males.
3. Goal-Setting/Self-Monitoring feature is more likely to motivate older people to use a fitness app than younger people.
4. Social Comparison feature is likely to motivate younger people to use a fitness app, but likely to demotivate older people.

Our main contribution to knowledge is as follows. First, our study is the first to demonstrate using a PLSPM approach that Goal-Setting/Self-Monitoring is the strongest driver of the *intention to use* a fitness app, regardless of gender and age. Second, in a prior study in the behavior theory domain, Oyibo et al. [35] found that, among individualist members, Perceived Self-Efficacy and Perceived Self-Regulation (mapped to Goal-Setting and Self-Monitoring) are the strongest drivers of physical activity. The authors also found that both social-cognitive determinants are stronger than Social Support (mapped to Competition, Cooperation, Social Comparison, etc.) for the individualist culture. In furtherance of research in the application domain, the current study showed that Goal-Setting/Self-Monitoring is the strongest driver of the *intention to use* a fitness app among members of the individualist culture. Moreover, we found that Competition and Cooperation are significant drivers of the *intention to use* a fitness app as well. Just as in the behavior theory domain, in the application domain, we found that Goal-Setting/Self-Monitoring is a stronger predictor of a fitness app's use than Competition and Cooperation among people living/residing in individualist cultures. These findings lay the groundwork for investigating in a field setting in the future whether Goal-Setting/Self-Monitoring will be more effective in motivating the physical activity behavior of individualist users than social features such as Competition and Cooperation.

*5.5. Limitations*

Our study has a number of limitations. The first limitation of our study is that it is not based on an actual fitness app. Rather, it is based on a prototyped fitness app, storyboards and users' intention to use a fitness app, the last of which does not imply the actual usage behavior. Thus, our findings may not generalize to an actual setting, in which a real-life application, equipped with the investigated features is used by the study participants to motivate their physical activity. The second limitation of our study is that most of the participants that took part in the study are citizens/residents of Canada and United States. This may threaten the generalizability of our findings to other populations outside Canada and United States. Therefore, in our future work, we intend to address these limitations by investigating the relationships between users' receptiveness to the investigated persuasive features and their intention to use a fitness app among non-Canadian/Americans or people residing in Canada and United States.

## 6. Conclusions

We have presented a path model to investigate the relationship between users' receptiveness to persuasive features and their intention to use a fitness app. The persuasive features were illustrated on storyboards as a case study. We found that Goal-Setting/Self-Monitoring is the strongest predictor of the *intention to use* a fitness app, followed by Reward, Competition and Cooperation. Based on our findings, we recommended that in a minimally viable fitness app, especially for users from individualist cultures, Goal-Setting/Self-Monitoring should be given priority to make the app appeal to a wider audience, especially to older people who are more motivated to use the fitness app based on this feature. In addition, in a personal setting, Reward should be supported by the fitness app. Moreover, at a social level, in addition to the personal features of Goal-Setting/Self-Monitoring and Reward, Competition, followed by Cooperation, should be supported. Comparatively, our results showed that the Cooperation feature is more likely to motivate females to use a fitness app than males. Moreover, Social Comparison is likely to motivate younger people to use a fitness app, but demotivate older people. As a result, Social Comparison should only be implemented in fitness apps targeted at younger people. In future research efforts, we look forward to investigating the generalizability of our findings to other populations outside Canada and United States.

**Author Contributions:** Data curation, K.O.; Funding acquisition, J.V.; Investigation, K.O.; Methodology, K.O.; Supervision, J.V.; Writing – review and editing, J.V. All authors have read and agreed to the published version of the manuscript.

**Funding:** This research was funded by the Natural Sciences and Engineering Research Council of Canada (NSERC) Discovery Grant (RGPIN-2016- 05762) of the second author.

**Conflicts of Interest:** The authors declare no conflict of interest. The funders had no role in the design of the study; in the collection, analyses, or interpretation of data; in the writing of the manuscript, or in the decision to publish the results.

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
