# Peer review of "Persuasive Features that Drive the Adoption of a Fitness Application and the Moderating Effect of Age and Gender"

_mti, doi:10.3390/mti4020017_

Round 1

Reviewer 1 Report

It’s a good paper that try to solve a good set of research questions.
Perhaps the author can change a little bit the structure and present objectives before methodology. The objectives aren’t part of the methodology. I’m sure that are a easy fixed problema.
The methodology and data analysis are exhaustive and it offer a good strategy to analyze this kind of datasets.
I appreciate the “tailoring” attemp analyzind differentialy by age,gender,etc. It must be a “must” in persuasive design.
I’m agree with limitations and I invite authors to use the same methodology with “real” mobile applications…taking a set of them that are “the best” (downloads, ratings, etc) and figure out what can happen.
In other hand….and it’s a general limitation….we can measure better “intention” that behavior….we have a problem because “intention” don’t imply, always, “behavior conversion”…but don’t affect the quality of this paper.

Author Response

It is attached.

Reviewer 2 Report

Overall:

This article addressed the interesting topic of which persuasive features used in fitness applications influence the user's intention to use such an application. The results of this study are relevant for designers of persuasive health applications because it gives insight in which features to use for which target user group. 

The design of the study is adequate, the results are presented in a clear way and the discussion and conclusions are interesting. I do have some concerns mainly related to the structure of the article and the relevance of the study, which I explain below.

If the proposed changes (mostly textual) are applied, I would accept the article for publication.   

Chapter 1 Introduction

The introduction presents detailed information about the results of the study. This information should be moved to results section, as it does not belong in the introduction. 

The introduction does not explain adequately what is the relevance of the research. The study sheds light on the relationship between persuasive features and intention to use. What does it add to for instance studies that addressed the influence of persuasive features on actual exercise behavior? What is the aim of this study? In chapter 5 design guidelines are presented, so apparently the authors aim to support designers of persuasive fitness applications. This should be adressed in the introduction.  

Chapter 2 Background and related work

Chapter 2 introduces the 6 persuasive features that the authors address in this research. However, the choice for these 6 features is not clearly justified based on literature. On which theories is this selection based? Authors could refer to for instance the PSD model or the Physical Activity Maintainance (PAM) model. This issue is addressed briefly in the Method section, but could be more extensive and should be moved to chapter 2. 

it should also be explained in chapter 2 why it is interesting to look at different age groups and gender. Is there evidence that younger and older people and male and female users have different preferences?

References to literature need to be checked. The order is not right and some references are double. 

Chapter 3 Method.

The design of the study in itself is adequate.

The description of 3.1 Research objective is ill-structured. The authors mix up the descirption of the objectives and research questions and the description of the research method (using storyboards). Please restructure.

Unclear if there were any inclusion or exclusion criteria for the participants of the study. 

Table 3 should be presented in a more structured way. 

Chapter 3 should include an Analysis paragraph explaining how data analysis has been carried out, both for the qunatitative and for the qualitative data. Such a paragraph should also include an explanation of the distinction between younger and older people at 34 (despite the fact that age had 5 different answer categories in the questionnaire).

Chapter 4. Results:

Overall this chapter presents a clear description of the results.

The qualitative data are presented in a very anecdotal manner, showing only a small selection of the remarks that were made by the participants and without any thematic analysis of the data. The same individual quotes are used in the discussion chapter as well.  

Chapter 5. Discussion:

In the discussion the results are summarized and put in perspectice. Qqualitatieve and quantitatieve data are compared and the insights are in de Discussie worden de resultaten toegelicht, worden de kwalitatieve en kwantitatieve data aan elkaar gekoppeld (maar alleen op basis van enkele quotes, er heeft geen echte analyse van de kwalitatieve data plaatsgevonden) en worden de inzichten getoetst aan de literatuur. 

Table 5 shows a clear and helpful summary of the results, should be moved to the Results section (chapter 4).

In 5.4 the authors state they replicated the findings of Oyibo that social support is a strong indicator of physical behavior. Consider to reformulate, because since the current study does not address the actual behavior, but people's intention to use a fitness application, this cannot be called not a replication.

Minor changes:

Title: fitness application -> a fitness application / fitness applications

s21/22: feature -> a feature / features

Table 1: the difference between social learning and social comparison is not clear from these definitions.

Table 2: 35-34 -> 35-44

s252: Incorrect sentence: For the respective submodels are similar to the global model.

s288: support -> supports

s289-292: do not use percentages to indicate GOF. 

s384: lower (higher) users -> lower (higher) older users

s401: explain SCT

s410: in one-size-fits-all fitness app -> add 'a' or make plural

s447: in the folds? 

Authors should check the references. The order in the text is not right, and some of the references appear in the reference list more than once, e.g. 3 and 4, 2 and 11.

Author Response

It is attached.
